# Exploring the potential of nest archives for establishing long-term trends in local populations of an Arctic-nesting colonial sea duck

Inmaculada Álvarez-Manzaneda[1,2]*, Kathleen M. Rühland[2], Marlo Campbell[2], Matthew P. Duda[2], Mark L. Mallory[3], Nik Clyde[4], H. Grant Gilchrist[4], Kathryn E. Hargan[5]*, John P. Smol[2]

1 Departamento de Ecología, Universidad de Granada, Granada, Spain, 2 Paleoecological Environmental Assessment and Research Lab (PEARL), Queen's University, Kingston, Ontario, Canada, 3 Biology Department, Acadia University, Wolfville, Nova Scotia, Canada, 4 Environment and Climate Change Canada, Ottawa, Ontario, Canada, 5 Department of Biology, Memorial University of Newfoundland, St. John's, Newfoundland and Labrador, Canada

* miams@ugr.es (IA-M); khargan@mun.ca (KEH)

**Citation:** Álvarez-Manzaneda I, Rühland KM, Campbell M, Duda MP, Mallory ML, Clyde N, et al. (2025) Exploring the potential of nest archives for establishing long-term trends in local populations of an Arctic-nesting colonial sea duck. PLoS One 20(10): e0332605. https://doi.org/10.1371/journal.pone.0332605

## Abstract

Tracking changes in seabird populations from remote Arctic regions using traditional monitoring techniques is financially and logistically challenging, leading to limited information on historical population trends. In this pilot study, we use a novel application of paleolimnological proxies to track environmental change using bird nests. Specifically, we examine long-term population dynamics of the Northern Common Eider (*Somateria mollissima borealis*), a philopatric sea duck. Eider nests from the Canadian sub-Arctic were sampled and radioisotopically dated, indicating that eiders have been nesting here since the 1800s. To assess the applicability of paleoecological proxies in nests to monitor environmental changes and long-term eider population dynamics, we examined changes in diatom species composition, shifts in the abundance of siliceous proxies (i.e., diatoms, chrysophyte cysts, phytoliths, protozoan plates), visible reflectance spectroscopy-inferred chlorophyll *a* (VRS-chla), stable nitrogen isotopes, and a selection of metal(loid)s. Warmer post-Little Ice Age conditions after the mid-19th century, together with higher eider occupation rates, promoted the proliferation of diatoms and other siliceous indicators. Declining eider populations during the industrial era, likely due to increased hunting pressures, was indicated by declines in $\delta^{15}N$ values and relative abundances of diatom taxa typically associated with higher nutrients and/or moisture. Increasing concentrations of metals (i.e., Zn and Cd), $\delta^{15}N$ values, and VRS-chla, which are positively associated with eider nesting activity, provided further support that eider numbers increased during the latter part of the 20th century. Our study shows that the accumulated vegetative and peat material from eider nests can provide a powerful tool to track historical bird

**Data availability statement:** All data, figures and scripts used in the manuscript are uploaded to to Zenodo. The DOI is: https://doi.org/10.5281/zenodo.15526799.

**Funding:** This research was funded by Environment and Climate Change Canada, the W. Garfield Weston Foundation, a Kenneth M. Molson Foundation grant, the Natural Sciences and Engineering Research Council of Canada (NSERC), Oceans North and the PEW Charitable Trusts, Nunavut General Monitoring Plan, ArcticNet Network Centres of Excellence, and was within the project PAST, financially supported by the European Commission (H2020-MSCA-IF-2019, Grant no. 897535).

**Competing interests:** The authors have declared that no competing interests exist.

population dynamics in ways that traditional, more recent, population monitoring methods cannot. Collectively, these methods can contribute insights to guide conservation decisions of this harvested species and other under-surveyed species.

## Introduction

Fluctuations in seabird populations are strongly influenced by changing environmental conditions as a consequence of both natural variation and anthropogenic stressors. However, long-term population monitoring data are scarce, and when they are available, they are often sporadic, particularly in remote, hard-to-reach locations such as the islands of Hudson Strait in the Canadian Arctic [1,2]. Comprehensive conservation strategies that address the effects of anthropogenic impacts on seabird populations require long-term perspectives [3,4] and yet are rarely considered in conservation biology studies because of an absence of historical information [5,6]. A developing approach to access these missing baseline (pre-industrial) conditions is through multi-proxy paleoecological studies [7]. A better understanding of past changes is key for managing future species populations by anticipating the consequences of human impacts and climatic stressors [6]. Such historical perspectives can strengthen management decisions on species conservation, maintenance and restoration [7].

Common eiders (*Somateria mollissima*) are a culturally and economically significant species of sea duck that nest throughout the circumpolar Arctic, although limited information on population trends exists [8,9]. Eiders are long-lived sea ducks that have adapted to minimize exposure to terrestrial predators by establishing nesting colonies on small offshore islands throughout the Arctic [10]. Adult females typically return to the same nesting islands each year and often use pre-existing nests. Islands occurring in Arctic environments are characteristically nutrient-limited, so nutrient inputs delivered onto the islands by eider ducks have the potential to directly increase soil development, plant growth, and water retention, earning them the title of 'ecosystem engineers' [2]. For example, nests are initially built on bare rock and then slowly, over many years and perhaps even centuries, the addition of nutrient-rich guano, feathers, and eggshells stimulate primary production. This in turn promotes the establishment and buildup of vegetation in a distinctive ring around the nest bowl [2].

Common eiders have long been a valued cultural and subsistence resource for Indigenous peoples in the Arctic since the Pre-Dorset period [11,12] and are harvested for their down feathers, eggs and meat [13]. During the post-industrial era, common eider populations faced numerous anthropogenic pressures with technological advances in hunting (i.e., shotgun firearm and ammunition) and travel (i.e., motorized boats), both of which increased hunting efficiency over traditional methods [1]. Previous studies have linked the documented decline in eider populations nesting in the Canadian sub-Arctic to increased hunting pressure on their wintering grounds in Greenland, following the introduction of shotguns starting ca. 1910, with accelerated usage during the mid-20th century [1,11]. Additionally, local common eider populations

nesting in Hudson Strait have been affected by several epidemic outbreaks including avian cholera in 2004 [14–16], avian influenza virus in 2011 [17] and possibly the most recent 2021 outbreak of avian influenza virus that circulated in North America [18,19].

In the eastern Canadian Arctic, sporadic, localized censuses have been conducted since the 1950s on different island clusters of the common eider's breeding range [20–22], but there have been no comprehensive surveys and very limited monitoring. Most available historical information that predates coastal surveys of nesting islands is provided by Inuit local ecological knowledge [16]. Therefore, very little is known about the early history by western scientists of these sea duck populations in Arctic Canada prior to the 1960s [20].

In recent years, paleolimnological approaches have been used to track long-term trends in seabird populations using a variety of environmental proxies preserved in sediment records from Arctic ponds and lakes that are proximal to colonial nesting grounds (e.g., [1,7,23–25]). For example, Hargan et al. [1] used sedimentary fecal lipophilic biomarkers (sterols) together with long-term trends in stable nitrogen isotopes ($\delta^{15}N$) to reconstruct fluctuations in common eider numbers over time at several Canadian Arctic breeding sites. However, even though these lake archives serve as repositories of historical data, a key caveat is that the sedimentary records integrate signals from multiple avian species that utilize the same aquatic environment. For example, the sedimentary fecal biomarkers of lake sediments were also influenced by small numbers of other migratory birds, like gulls and geese [1]. Sampling nests has the potential to effectively obtain direct records from a single species.

In this study, we explore a novel application of paleoecological techniques to conservation biology that applies directly to accumulated nest material that had been deposited vertically. This is possible because common eiders often form a nest bowl around a depression in the ground by drawing in nearby plant material with their bill as well as by plucking feathers from their chest. Nesting material is gathered around the eider, shaping it to a bow [10]. Over time and many generations, this material can build up to form a substantial depression with an organic perimeter (see Fig 1). We sampled

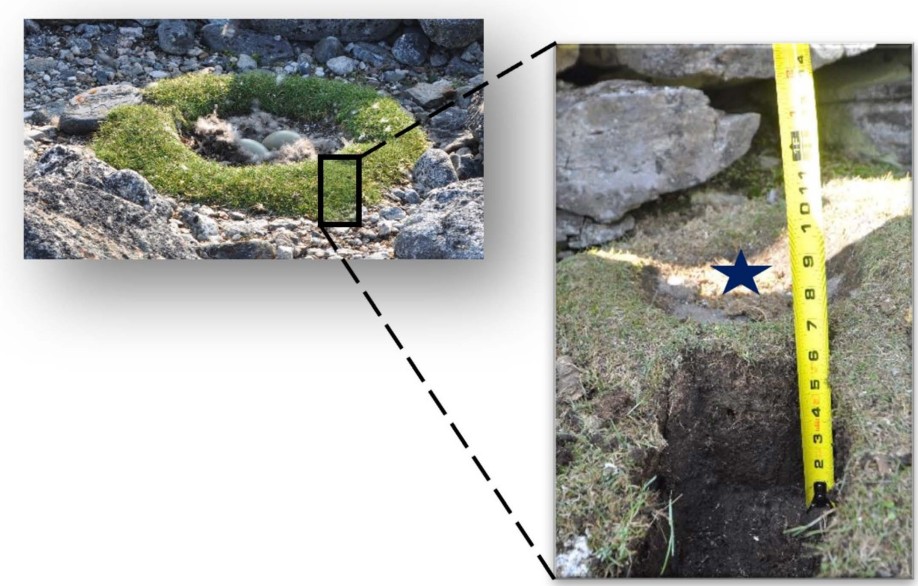

**Fig 1. An example of an eider nest from Digges Sound.** A vegetation ring around the eider nest cup (left; Credit: G. Gilchrist) and a vertical sequence of the material sampled (right; Credit: K. Hargan). The blue star marks the nest cup.

eider nests (the vegetation surrounding the nest bowl) from remote islands on Digges Sound in the Canadian Arctic where accessibility for survey monitoring is challenging and data remain scarce. Using a suite of environmental indicators preserved in the nest material, including diatoms and other siliceous proxies, isotopes, and metal(loids), we evaluate the potential for this approach to track environmental change and long-term population trends. Given the exploratory nature of this pilot study, we aim to evaluate the potential of the following: (i) whether paleolimnological approaches applied to eider nests can provide reliable geochronological and paleoecological archives; (ii) whether baseline conditions can be established to determine the duration of common eider occupation in these nesting grounds; and (iii) whether long-term changes can be tracked in the nest environment. The answers to these questions can provide a temporal context for better understanding current and future seabird population dynamics in this remote Arctic region.

### Digges Sound: Site description and brief history of the region

Eider nests were sampled in July 2014 from four islands in Digges Sound in the eastern Canadian Arctic (Hudson Strait–Northern Hudson Bay Narrows, Quebec) (Fig 2). Help accessing sites was received from community members in Ivujivik. Digges Sound is categorized as Low Arctic and forms an arm of deep water running southwest from Hudson Strait [26]. Offshore waters are typically covered in mobile pack-ice from January to April [27]; however, recent studies identify a decline in the duration of sea ice in the area [28–31]. The local community of Ivujivik (~420 people) has a long history of harvesting seabird colonies and sea duck nesting islands [27] and the harvest of marine birds continues to be an important part of its culture and food security today. Local hunting pressure of adult eider ducks is considered to be low to moderate, particularly in comparison to Greenland and Atlantic Canada where these sea ducks often overwinter [1].

Eiders are the predominant birds breeding on most of the small, low-lying islands south and west of Digges Sound [24,27]. The remoteness of these islands may also be an incentive for eiders to establish nests given that they are sensitive to human activities [13]. Female common eiders typically return to the same nesting island in successive years and typically lay their eggs in pre-existing nest bowls. In this open environment, hens have been found to preferentially select nest bowls that contain feathers and/or loose vegetation compared to nests with little or no nesting material [32]. Pre-existing nesting material has also been found to encourage earlier onset of nest initiation [32], and may offer numerous survival benefits to the clutch including providing exposed eggs with insulation, as well as concealment from avian predators. This is particularly important for the survival of first laid eggs, because they are left unattended while the hen leaves on foraging trips to accrue the energy necessary for clutch completion [32]. Given the reproductive advantages of existing nest bowls and the materials found within them, eider hens rarely lay eggs in an entirely new nest on exposed ground [2,10,32].

## Materials and methods

### Sample collection

A permit to Environment and Climate Change Canada (ECCC) was received from the Makivik Corporation 'Permit for Access and Entry – Nunavik Inuit Lands' through governance of the Nunavik Marine Region Impact Review Board. Furthermore, a permit to conduct scientific research on common eiders was issued under the Section 19 of the Migratory Birds Regulations to ECCC (NUN-SCI-12–02).

Samples of eider nests were collected from 5 nests, from four remote nesting islands in Digges Sound, Hudson Strait, Nunavik, Quebec, Canada (Fig 2). Data collection took place toward the end of the breeding season, between July 25–30, 2014. Nest samples were labelled as DS (Digges Sound) followed by an island ID and nest number ID, in cases where more than one nest was sampled on each island (e.g., E1 = eider island; N1 = nest 1). The nest profiles collected for this study include DS-E1-N1, DS-E1-N2, DS-E2-N1, DS-E3-N1 and DS-E5-N1. The five nests were selected based on their

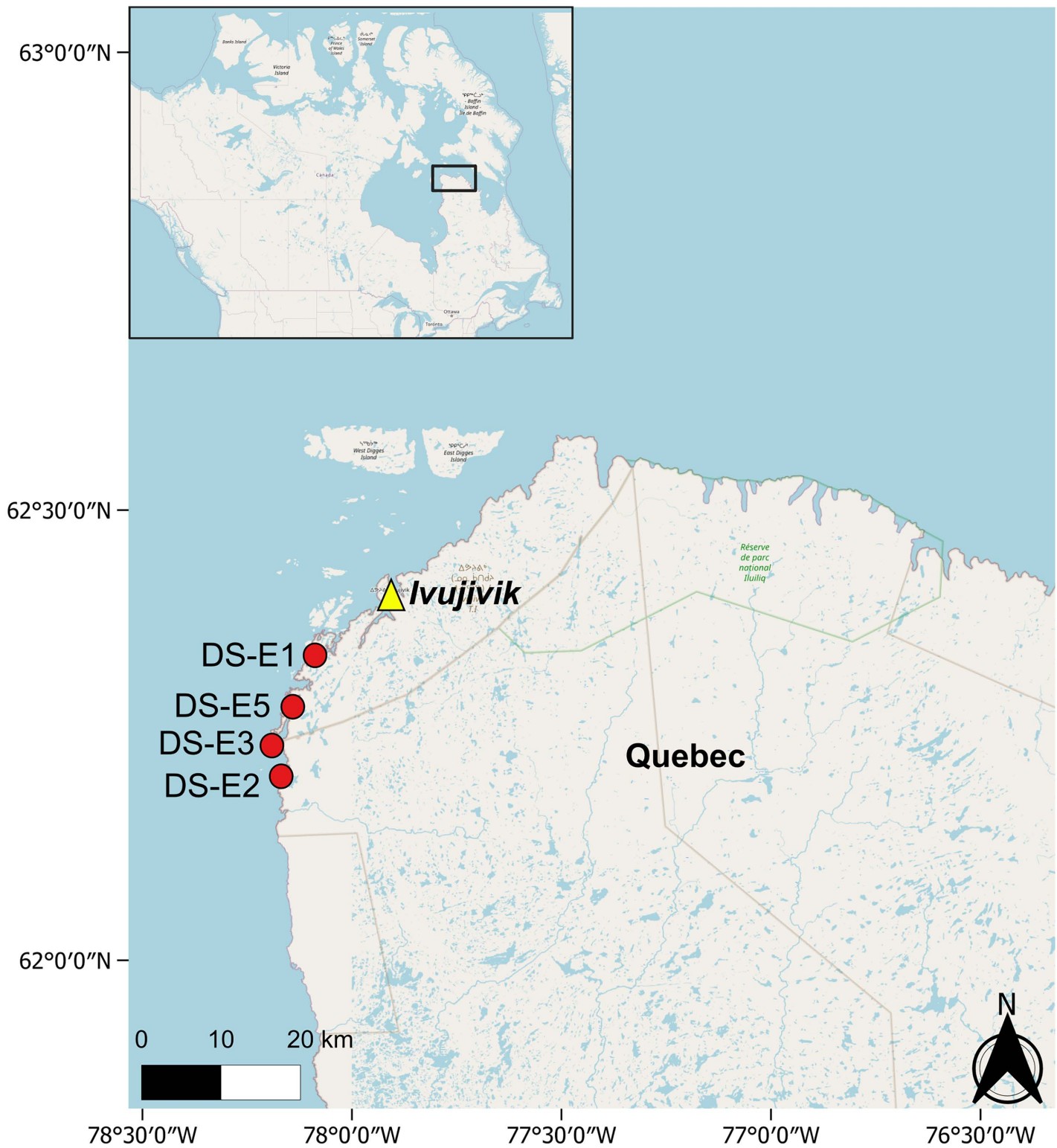

**Fig 2. Location of the eider nesting islands in Digges Sound, southwest of Ivujivik (islands are marked with red dots) where our samples were collected.** Base map and data from OpenStreetMap and OpenStreetMap Foundation (CC-BY-SA). Map rendered using QGIS.

proximity to a freshwater pond and the absence of eggs at the time of sampling (i.e., either unused in 2014 or depredated). Since this is a pilot study, we kept the sample size small to minimize potential habitat disturbance and resource use on the limited number of nests on the island that met our selection criteria. The vegetation ring that surrounds the nest cup (Fig 1) was sampled with a breadknife to retrieve a vertical section down to bare rock. The sample was then placed on a sheet of aluminum foil and sectioned into 1-cm contiguous intervals before being transferred into Whirl-Pak® bags and placed in a cooler. The hole left by removal of the sample was filled in with nearby moss that was not part of a nest. In addition, eider guano was collected with the aim to analyze its composition. Samples were transported to the Paleo-ecological Environmental Assessment and Research Laboratory (PEARL), Queen's University, Kingston, Ontario, Canada where they were stored in a cold room at ~4°C until ready for analyses, following general recommendations [33].

### Radioisotopic dating ($^{210}$Pb and $^{14}$C)

The nest profiles were analysed at PEARL using $^{210}$Pb gamma spectroscopy to estimate the accumulation of nest material over a period of 100–150 years, following standard methods [34]. For each nest profile, the aim was to analyse all available samples within the unsupported $^{210}$Pb inventory and at least three intervals in the supported $^{210}$Pb (background) part of the profile (between 10 and 15 samples). Each interval analysed used approximately 1.5 g of freeze-dried nest material, which was inserted into plastic gamma tubes to a height of ~3 cm and then sealed with 2-ton epoxy. Nest material with substantial vegetation was pulverized before its placement into tubes. Samples were left to rest for at least two weeks to ensure that $^{214}$Bi and $^{226}$Ra reached equilibrium. Activities of $^{210}$Pb, $^{137}$Cs, and $^{214}$Pb (proxy for supported/background $^{210}$Pb) were measured with an Ortec high-purity Germanium Gamma Spectrometer. Unsupported $^{210}$Pb activities were determined by subtracting supported $^{210}$Pb (i.e., $^{214}$Pb) from total $^{210}$Pb activities. The chronologies for the nest profiles were based on estimates of unsupported $^{210}$Pb activities and the constant rate of supply (CRS) model [35], using the ScienTissiME dating software. The http://www.scientissime.net/softwareCRS model is widely used for dating recent sediment profiles [34,35] and assumes a constant supply of unsupported $^{210}$Pb deposited from the atmosphere to the surface of vertically aggrading profiles, while allowing for variations in accumulation rates [35]. This model is particularly suitable for ombrotrophic systems that depend solely on atmospheric inputs, such as undisturbed peat deposits [36], and likewise eider nest profiles.

To obtain dates for the bottom intervals (i.e., when nest accumulation started on bare bedrock) of nests with longer profiles required a chronological method that is able to date materials that are older than the $^{210}$Pb method is capable of (i.e., for dating material below the $^{210}$Pb equilibrium depth). Radiocarbon ($^{14}$C) methods are widely used to date materials accumulated from hundreds of years to tens of thousands of years [37] and can complement $^{210}$Pb techniques used for more recent sediments [38]. For DS-E5-N1 and DS-E3-N1, peat macrofossils from one bottom sample were sent to the University of Ottawa, André E. Lalonde Accelerator Mass Spectrometry (AMS) Laboratory for radiocarbon analysis. Because these samples were partially humified, identification was not possible and bulk samples were used. Samples underwent an Acid-Alkali-Acid pre-treatment (see [39]). For each nest, age-depth modeling was performed in R for Windows version 4.2.1 [40], using $^{210}$Pb and $^{14}$C dates with the function 'clam' in the R package 'clam' v. 2.5.0 [41]. Radiocarbon dates were calibrated using the 'IntCal20' calibration curve [42].

### Diatoms and other siliceous indicators

The four siliceous indicators examined in this study provided a range of ecological information that allowed us to track environmental conditions both within the nest and the surrounding environment. In particular, the availability of moisture and nutrients required for the establishment and persistence of diatoms, chrysophytes and protozoa within the nest as well as phytoliths in the surrounding grasses is linked to the presence of nesting eiders. Diatoms are particularly responsive to variations in nutrient and moisture levels and shifts in diatom species composition over time can track changes in these conditions. Chrysophycean algae are often associated with mosses and their stomatocysts (resting cysts) may provide additional information on changes in moisture levels [43]. Testate amoebae (protozoa) are commonly found in the

moist environments of peat bogs [44], and changes in the relative abundance of their siliceous tests (plates) can provide additional information on fluctuating moisture levels within the nest environment. Phytoliths, which are siliceous deposits in plants (mainly grasses and sedges), are released directly into the soil upon decomposition of the plant tissue [45]. Therefore, changes in the relative abundances of phytoliths in the nests can provide insights into the surrounding grass and sedge vegetation.

Diatom samples were analyzed for all available intervals of each nest profile, generally following methods described in Rühland and Smol [46]. In addition, guano samples were also analyzed for diatom content using the same procedures. In brief, ~ 0.1 g of freeze-dried nest material from each interval was placed into glass scintillation vials and then treated with a 1:1 molecular weight ratio of concentrated sulfuric acid ($H_2SO_4$) and concentrated nitric acid ($HNO_3$) to digest the organic matrix. The samples were then placed in a hot water bath and heated to 80°C for at least 2 h. Samples were allowed to settle for 24 h, after wich the supernatant was removed and replaced with distilled water. This rinsing procedure was repeated (allowing the diatoms to settle completely between rinses) until a litmus test indicated that the samples were no longer acidic (~6 times). For each interval, aliquots of the slurry were pipetted onto glass coverslips in four dilutions and allowed to dry before being mounted onto microscope slides using Naphrax®.

Diatom enumeration was undertaken using a Leica DMRB microscope under oil immersion at 1000 × magnification and fitted with differential interference contrast optics. Where feasible, at least 400 diatom valves were counted in each interval [47,48]. In intervals where diatoms were less plentiful (the second interval of DS-E1-N2 and the last interval of DS-E3-N1), at least 200 valves were counted. Counting effort was deemed impractical and too low for reliable calculations of percent relative abundances, when only a few valves were encountered in ~100 fields of view. Diatoms were identified to the lowest taxonomic level possible (often variety) using widely accepted taxonomic sources [49–52] and AlgaeBase [53] to update to currently accepted taxonomic names. For each interval, diatom counting data were expressed as a percent relative abundance of the total number of diatom valves counted. Using the same slides prepared for diatom enumeration, chrysophyte cysts, protozoan plates, and phytoliths were also counted (but not taxonomically identified). As noted above, chrysophyte cysts and protozoan plates serve as indicators of environmental moisture, whereas phytoliths can provide insights into surrounding grass vegetation and nest material. An index for each siliceous indicator was expressed as a percentage of the sum of all four siliceous indicators (i.e., diatoms, chrysophyte cysts, protozoan plates, phytoliths). For example, an index of chrysophyte cysts was calculated using the formula: (#cysts/(#cysts + sum of diatom valves, phytoliths, protozoan plates))*100. In samples where cysts, protozoan plates or phytoliths were plentiful, these indices were based on the first 100 diatom valves enumerated. The percent relative abundances of the most common diatom taxa and the relative frequencies of the four siliceous microfossils were displayed stratigraphically using TGView version 2.6.1 [54]. Significant biostratigraphic zones were determined for diatoms by a broken-stick model in R for Windows version 4.2.1 [40], with the packages 'ggplot2' v. 3.5.1 [55] and 'tidypaleo" v. 0.1.3 [56], after performing constrained incremental sum of squares (CONISS) cluster analysis with the chord distance as the dissimilarity coefficient. CONISS is widely used in paleoecology to divide biostratigraphic sequences into zones of compositional change over time [57], whereas the optimal number of meaningful zones can subsequently be determined by applying the broken-stick method [58].

## Geochemical analyses

Freeze-dried material from selected intervals spaced throughout each nest profile were analysed for a suite of metal(loid)s at Société Générale de Surveillance (SGS) in Lakefield, Ontario, Canada, using *aqua regia* digestion (which extracts metals while preserving the silicate matrix) and inductively coupled-plasma mass spectrometry (ICP-MS). Thirty-two elements were analysed, but we focused on three elements that have been related to eider inputs, Zn, Cd, and Pb [59]. For example, seabirds generally accumulate high concentrations of trace elements due to their bioaccumulation and

biomagnification throughout the food web [60]. Eider ducks specifically bioaccumulate Zn, Cd and Pb as they consume mollusks and benthic crustaceans which are enriched in these elements [59–61]. Temporal fluctuations in stable Pb concentrations in each profile were plotted against historical accounts of shotgun sales from the west coast of Greenland [62] to examine whether changes in hunting practices (i.e., use of lead shot) and industrial heavy metal contamination can be detected in the eider nest records. Historical data (Christian Vibes annotations, Arctic Institute archives) for rifles and shotguns that were sold or traded to Greenlandic hunters were downloaded from [62].

### Chlorophyll *a*

Trends in visible reflectance spectroscopy-inferred chlorophyll *a* (VRS-chla) concentrations [63] were examined in an exploratory fashion for each dated eider nest profile as a potential link to increased nutrients driving higher primary production from eider occupation. For each sample, freeze-dried nest material was sieved through a 125-µm mesh to homogenize particle size prior to being placed into glass cuvettes and scanned with a Model 6500 series Rapid Content Analyzer (FOSS NIRSystems Inc.). VRS-chla data were analysed using the log-transformed data from [63] with spectral absorbances of wavelengths between 650 and 700 nm, including the primary degradation products (pheophytin *a*, pheophorbide *a*) as they have similar spectral signatures [64].

### Stable nitrogen isotopes

All samples were measured for elemental composition (%CN) and stable nitrogen isotopes ($\delta^{15}N$) at the Earth Resources Research and Analysis (TERRA) facility at Memorial University. Approximately 8–9 mg of freeze-dried material (depending on the organic matter content and C:N) was weighed into tin capsules. Analyses were conducted using a Carlos Erba elemental analyser coupled to a Delta V Plus isotope ratio mass spectrometer (EA-IRMS, Thermo Scientific) via a Con-Flo III interface. $\delta^{15}N$ results were reported using delta ($\delta$) notation in parts per thousand (‰) enrichments or depletions relative to AIR (atmospheric nitrogen). The data were normalized by using previously calibrated internal standards, with an analytical precision of ±0.2%.

Zn, Cd, Pb, VRS-chla and $\delta^{15}N$ results were summarized by using Z-scores (standardized within nest profiles to a mean of 0 and a standard deviation of 1) to ensure comparability across nest profiles and proxies. Major patterns of variation in diatom assemblages and relative abundances of other siliceous indicators were summarized by principal component analysis (PCA) in R for Windows version 4.2.1 [40], using the package, 'vegan' v. 2.6−4 [65]. To summarize the main patterns of variation over time, PCA axis 1 sample scores based on relative abundances of diatom taxa (PCA-1$_{diatoms}$) and PCA axis 1 sample scores of the relative abundances of the four siliceous indicators (PCA-1$_{indicators}$) from each nest were plotted against time.

## Results

### Radioisotopic dating ($^{210}$Pb and $^{14}$C)

Total $^{210}$Pb activity generally followed a characteristic decline with depth in four of the five eider nest profiles (Fig 3). However, DS-E1-N2 and DS-E3-N1 activities peaked below the surface interval, before following a typical decline with depth. Initial concentrations of total $^{210}$Pb activities ranged from 1251 Bq kg$^{-1}$ to 368 Bq kg$^{-1}$ before reaching equilibrium with $^{214}$Pb (i.e., supported $^{210}$Pb). For DS-E1-N1, the shortest nest profile collected (4.5 cm), $^{210}$Pb activity remained relatively high, and did not undergo an exponential decline between top and bottom intervals (Fig 3e): this nest was not analyzed further. For the remaining four nest profiles, equilibrium/background was reached between 5.5 to 11.5 cm (Fig 3). In general, no clear $^{137}$Cs peaks were detected in the nest profiles.

Establishing chronologies for the nest profiles DS-E5-N1 and DS-E3-N1 entailed combining $^{210}$Pb and $^{14}$C dating results, as the bottom intervals of these nests lie beyond the maximum range possible for obtaining reliable dates using $^{210}$Pb

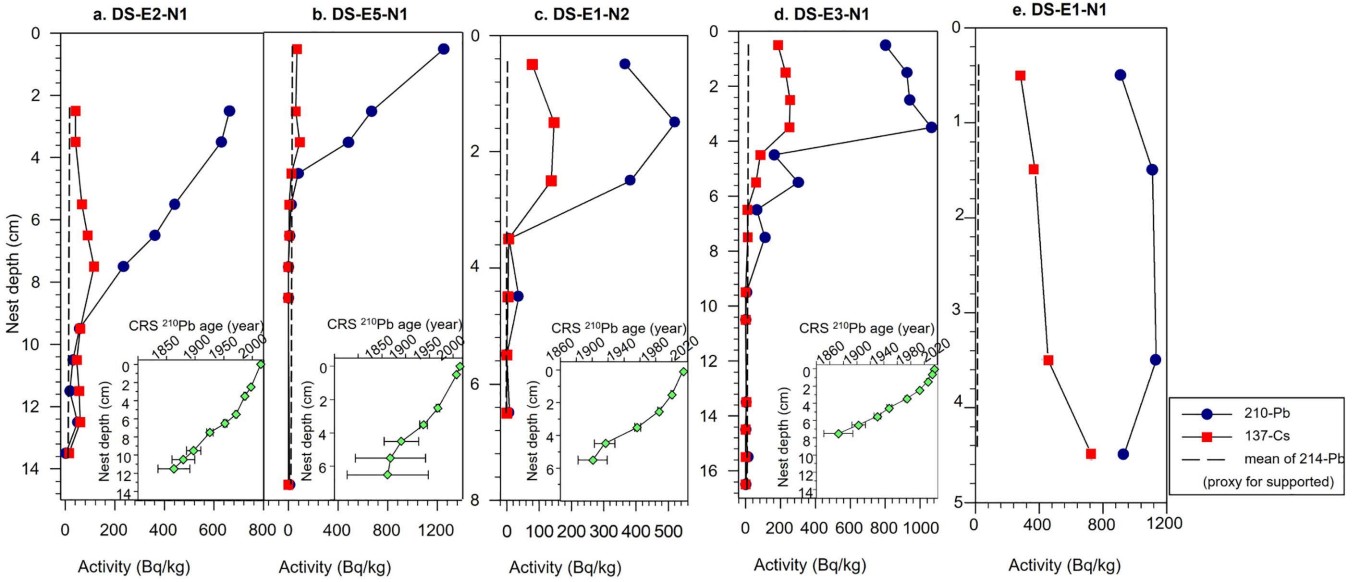

**Fig 3. $^{210}$Pb and $^{137}$Cs activity trends and the estimated dates based on the constant rate of supply (CRS) model plotted against nest depth (inset bottom right corners) for each Digges Sound nest profile.** The dashed vertical line represents the mean of $^{214}$Pb.

(~150 years) methods but were within the upper range of $^{14}$C methods. Radiocarbon dating of the bottom interval of nest profile DS-E5-N1 (13–14 cm) yielded an estimated age of ca. 1240±23 CE (Common Era) (733–688 cal BP, calibrated years Before Present) (see S1 Table). Although 16–17 cm was the bottom interval of DS-E3-N1, we opted to date the interval 11–12 cm as nest samples in lower intervals contained high levels of rocky material (i.e., *ex situ*) and therefore not ideal for establishing a radiocarbon date in this nest sample. Calibrating the DS-E3-N1 $^{14}$C date using IntCal20 proved challenging as would be expected with material younger than 500 years and yielded multiple age possibilities (S1 Table). The age-depth modeling of DS-E3-N1, performed using the R package 'clam' v 2.5.0 [41], returned a best fit date of 153 cal BP (~1797 CE), and allowed interpolation between the $^{210}$Pb chronology and $^{14}$C data.

### Trends in diatoms and other siliceous indicators

The four Digges Sound eider nest profiles archived the remains of diatom taxa and other siliceous indicators (chrysophyte cysts, phytoliths, protozoan plates) (Figs 4–7). The eider nest diatom assemblages were generally species-poor (~12–20 taxa), with the exception of DS-E5-N1 where 41 taxa were recorded. In the nest records DS-E1-N2 and DS-E3-N1, intervals below ca. 1985 and ca. 1904, respectively, did not yield reliable counts as the remains of all siliceous indicators were rarely observed (Figs 4,6). In these intervals, there was no evidence of diatom dissolution: the rare diatom valves that were encountered were well preserved and thickly silicified chrysophycean stomatocysts that would be less prone to dissolution, were also present.

We aimed to determine whether the diatoms observed in the eider nest material were derived from within the nest, were blown in from the nearby marine environment, or originated from nearby ponds that eiders drink from and thereafter defecate into the nest. Marine diatoms were not observed in the nest material, and therefore sea spray can be excluded as a potential source for the nest diatoms. In fact, diatom assemblages in all four eider nest profiles were characterized by aerophilic taxa that are frequently associated with terrestrial or ephemeral environments (i.e., able to tolerate periods of dry conditions). For example, *Pinnularia borealis*, *P. intermedia* and *P. divergentissima* were observed in high relative abundances in all four nest profiles (Figs 4–7) together with *Luticola mutica, Hantzschia amphioxys, Cosmioneis pusilla*,

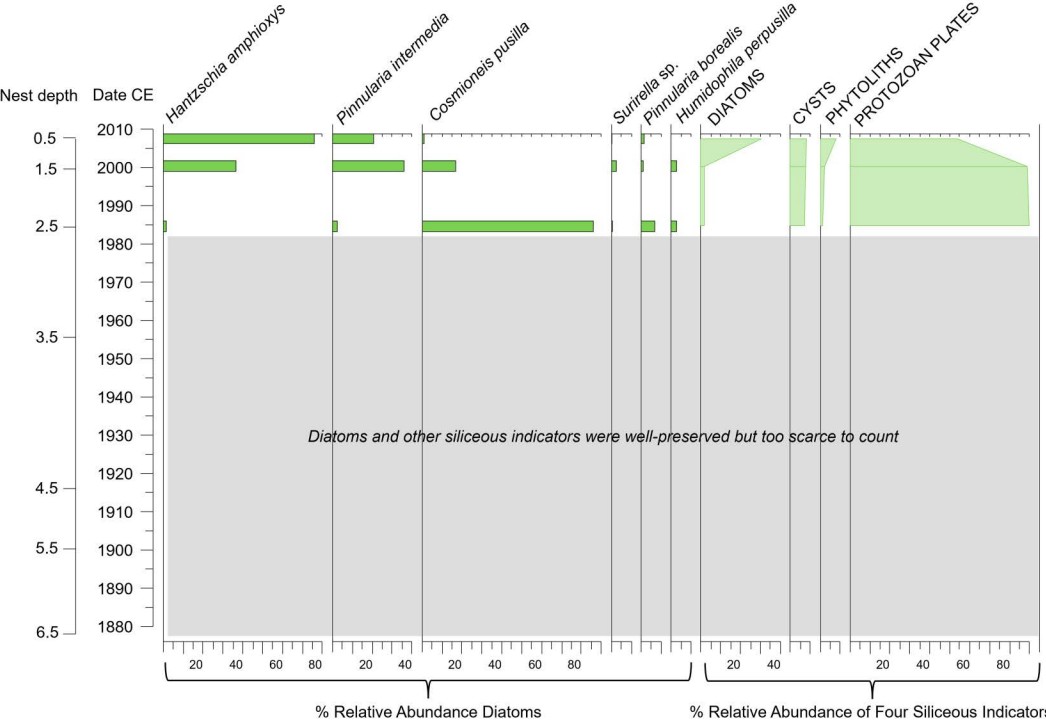

**Fig 4. Diatom relative abundances of the most common taxa encountered (solid bars on left) together with the percent relative abundance of each of the other siliceous indicators used in this study relative to the sum of all four indicators (silhouette lines on right) in the Digges Sound eider nest DS-E1-N2.**

*Mayamaea atomus*, *Humidophila perpusilla* and *P. sinistra* (additional information in S2 Table). In addition, diatoms did not originate from eider guano as no valves were observed in guano samples collected for this study. Relative abundance fluctuations among the four siliceous proxies varied across nest profiles, with protozoan plates being the dominant indicator in all records (ranging from 40–90% of the abundances) (Figs 4–7). Proportions of chrysophyte cysts were generally low (<10%) with the exception of nest DS-E2-N1 where this indicator reached 40% relative abundance in the earlier intervals (before ca. 1952). Phytoliths were generally abundant in all nest profiles, with a notable increase ca. 1980 in DS-E2-N1 (Fig 5).

**Temporal trends of diatoms by nest.** *DS-E1-N2* (Fig 4): Only the top three samples of the record (ca. 1985 to ca. 2014) contained countable diatoms (and other siliceous indicators) for assessment. Assemblages in this nest record were species-poor, with 12 taxa. The diatom assemblage at the bottom of the record (ca. 1985) was dominated by *C. pusilla* (~80% relative abundance). Protozoan plates were by far the dominant siliceous indicator in this interval (90%). *H. amphioxys* (~37% relative abundance) and *P. intermedia* (~36%) abruptly increased from trace abundances ca. 2000 to top of diatom record (ca. 2014) replacing the previously dominant *C. pusilla*. This shift in diatom composition around 2000 was concurrent with a pronounced decline in protozoan plate relative abundances and an increase in diatoms and the phytoliths (Fig 4). Chrysophyte cyst relative abundances remained stable and low throughout the short nest record.

*DS-E2-N1* (Fig 5): Diatom assemblages in nest DS-E2-N1 were relatively diverse compared to other eider nest records from this study, with diatoms occurring in countable abundances throughout the profile. Using CONISS and broken stick analysis, to identify the main shifts in biological proxy composition over time and determine the optimal number of meaningful zones, we identified four significant diatom zones. Within Zone 1 (ca. 1850 to ca. 1890)

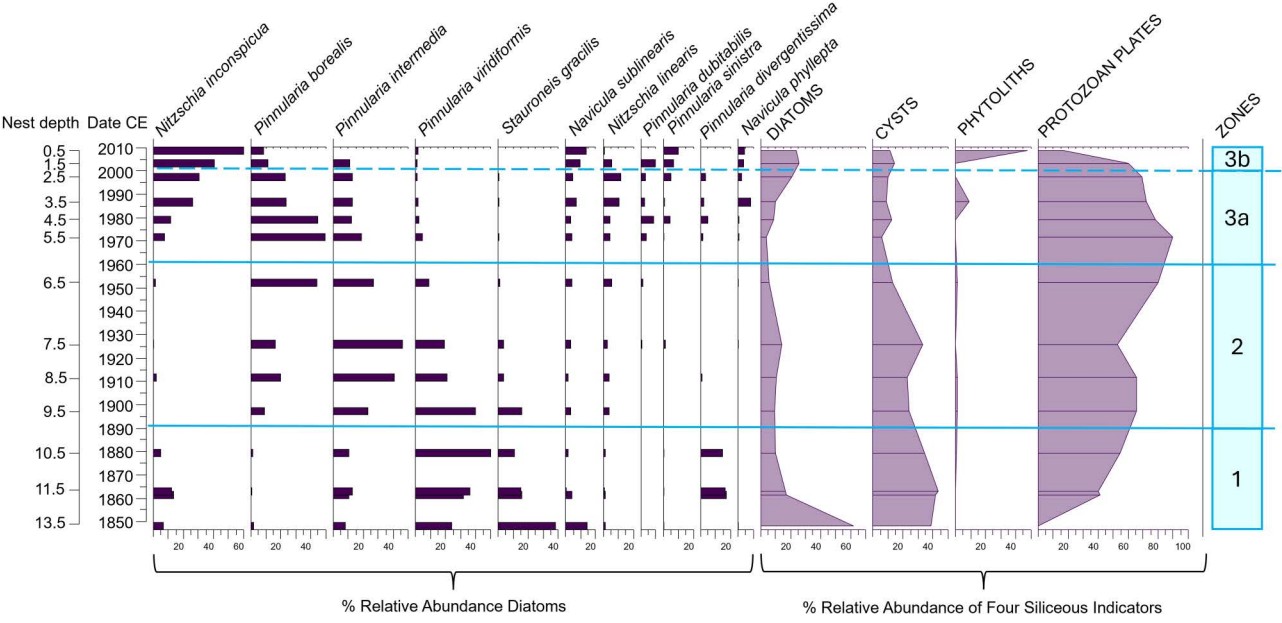

**Fig 5. Diatom relative abundances of the most common taxa encountered (solid bars on left) together with the percent relative abundance of each of the other siliceous indicators used in this study relative to the sum of all four indicators (silhouette lines on right) in the Digges Sound eider nest DS-E2-N1.**

*Stauroneis gracilis, P. viridiformis, Navicula sublinearis, Nitzschia inconspicua, P. intermedia* and *P. divergentissima* characterized the assemblage, with minor fluctuations among these taxa. Most notably, *S. gracilis* and *N. sublinearis* declined towards the end of this zone, whereas *N. inconspicua* and *P. divergentissima* briefly increased above trace abundances (12% and 17%, respectively) in the middle of this zone (ca. 1865 to ca. 1880), thereafter returning to trace abundances. Diatoms, chrysophyte cysts, and protozoan plates were abundant and underwent minor fluctuations during this zone, whereas phytoliths occurred in trace abundances (Fig 5). The transition to Zone 2 (ca. 1890) was characterized by a notable increase in *P. intermedia* together with an increase in *P. borealis* above trace abundances and equally pronounced declines in *P. viridiformis* and *S. gracilis*. Of the four siliceous indicators, protozoan plates and chrysophyte cysts were most prevalent. In Zone 3 (ca. 1960), *P. borealis* continued to increase, replacing *P. intermedia* and *P. viridifomis*. There was a marked shift in composition at ca. 1960 with assemblages that were previously dominated by *P. borealis* and *P. intermedia*, typical from nutrient-rich and drier conditions, now replaced by a variety of taxa, more associated with wetter habitats, that previously occurred in trace abundances including *Nitzschia inconspicua, N. sublinearis, N. linearis, P. dubitabilis, P. sinistra, P. divergentissima* and *Navicula phyllepta*. Subzone 3b was mainly characterized by the notable increase of *N. inconspicua* and *N. sublinearis* together with the disappearance of *P. intermedia* and very low abundances of *P. borealis*. The four siliceous indicators also experienced pronounced shifts in relative abundance, most notably with a rise in phytoliths above trace abundances for the first in the record. This was followed by a further increase in phytoliths in subzone 3b, together with an increase in diatoms and declines in chrysophyte cysts and protozoan plates (Fig 5).

*DS-E3-N1* (Fig 6): Diatoms and other siliceous indicators were rarely observed in the early parts of the nest profile and did not become established until ca. 1900. For the remainder of the record, diatom assemblages underwent minor fluctuations that were not deemed significant by broken stick analysis. *P. borealis* was the dominant taxon (mean relative abundance = 54%) together with *M. atomus* that reached a maximum relative abundance of 32% at ca. 1930. Other aerophilic taxa observed in lower abundances included *P. intermedia, P. sinistra, H. amphioxys, L. mutica* and *P. minutiformis,* the

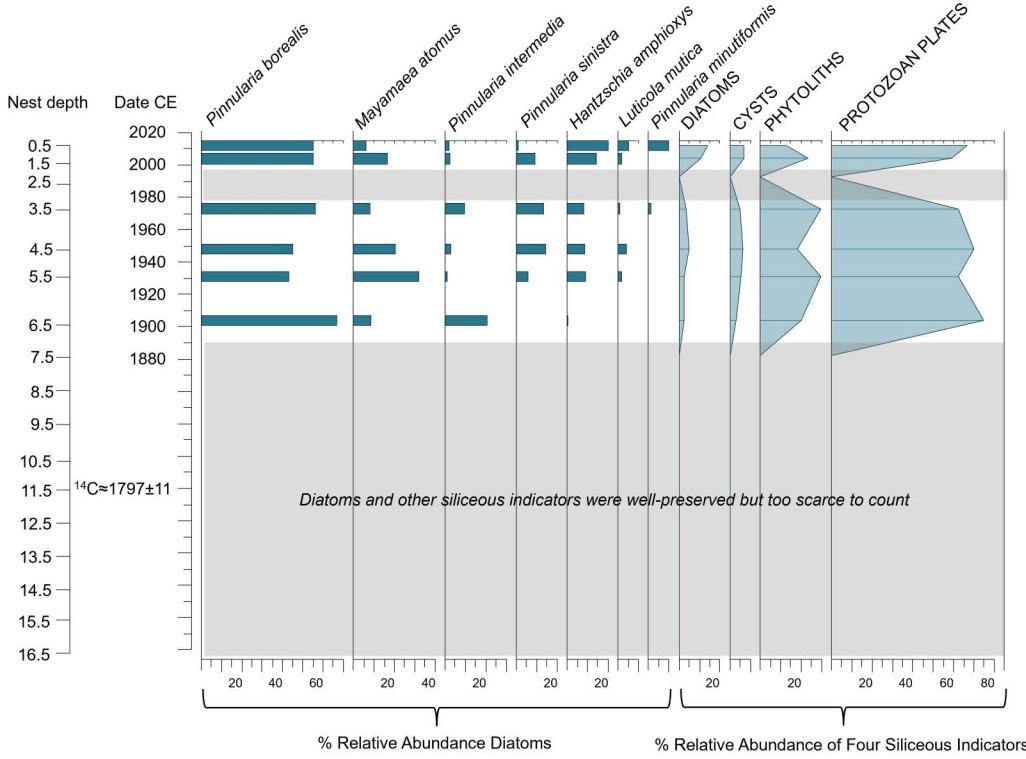

**Fig 6. Diatom relative abundances of the most common taxa encountered (solid bars on left) together with the percent relative abundance of each of the other siliceous indicators used in this study relative to the sum of all four indicators (silhouette lines on right) in the Digges Sound eider nest DS-E3-N1.**

latter three increasing slightly in the most recent intervals. Relative to phytoliths and protozoan plates, chrysophyte cysts and diatoms were scarce in this nest profile.

*DS-E5-N1* (Fig 7): Diatoms were observed in countable abundances throughout the nest profile. Using CONISS and broken stick analysis, we identified three significant zones. Diatom assemblages were dominated by *P. intermedia* (mean relative abundance = 71%) for the entire sequence. At the earliest stage of nest development (ca. 1240 ± 23 CE), *P. intermedia, L. mutica* and *M. atomus* characterized the assemblage, with minor fluctuations among these taxa. Other aerophilic taxa occurred in trace abundances including *Cocconeis costata, C. pusilla* and *P. borealis*. Phytoliths and protozoan plates had higher abundances (33–58%) in comparison with diatoms and chrysophyte cysts (<10%) (Fig 7). The transition to Zone 2 (ca. 1600) was characterized by significant decreases in the relative abundances of *L. muticola* and *M. atomus*, along with an overall increase in *P. intermedia*. Diatoms and cysts remained in trace abundances, whereas protozoan plates increased in relative abundance while phytoliths declined around after ca. 1300 followed by another shift after ca. 1350. Zone 3 (starting ca. 1800) was characterized by the rise (albeit in relatively low abundances) in a variety of aerophilic diatom taxa that were previously not observed above trace abundances in the nest profile including *P. borealis, P. minutiformis*, and *P. sinistra* as well as a return to higher relative abundances of *M. atomus* and a decrease in *L. mutica* in subzone 3b (ca. 1950). Protozoan plates remained the most abundant siliceous indicator followed by phytoliths, whereas diatoms and chrysophyte cysts remained in low relative abundances although diatoms increased in the top-most interval, reaching maximum abundances (~15%).

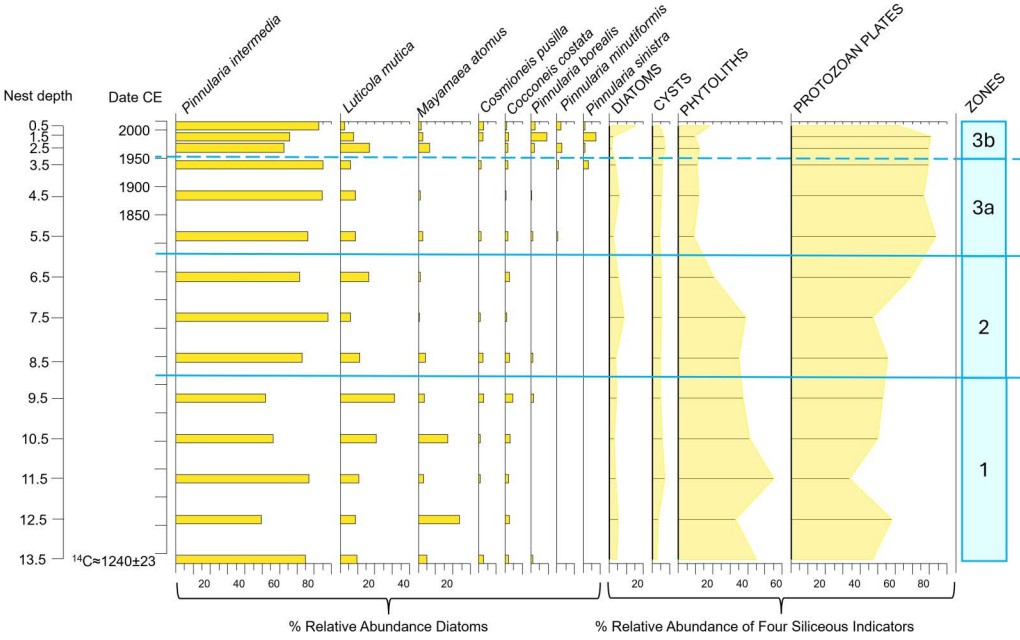

**Fig 7. Diatom relative abundances of the most common taxa encountered (solid bars on left) together with the percent relative abundance of each of the other siliceous indicators used in this study relative to the sum of all four indicators (silhouette lines on right) in the Digges Sound eider nest DS-E5-N1.**

## Temporal trends in ornithogenic proxies from nest profiles

To better track changes in eider populations over time, concentrations of Zn, Cd and Pb were represented together in Fig 8 with other ornithogenic variables, since they have been shown to have strong independent correlations to seabird inputs [59]. The seven ornithogenic proxies examined in the dated eider nest records (Zn, Cd, Pb, $\delta^{15}N$, VRS-chla, PCA-1$_{diatoms}$, and PCA-1$_{indicators}$) fluctuated, although not always similarly across nests (Figs 8 and S1). In general, Zn, Cd and VRS-chla concentrations increased after the mid-20th century but timing varied across nest profiles. For example, Zn and Cd increased ca. 1925 in DS-E2-N1, Zn in ca.1950 (Cd initially ca. 1840 and more pronounced ca. 1950) in DS-E3-N1, ca. 1970 for both Zn and Cd in DS-E5-N1, and ca. 1985 for both Zn and Cd in DS-E1-N2 (Fig 8).

Although not concurrent across nest profiles, Pb concentrations generally increased starting ca. 1850, declining from the early to mid-20th century to the present (Fig 9). The exception to these trends was nest DS-E1-N2, where Pb concentrations underwent a further increase after ca. 1960, then declining only after ca. 2000.

Trends in $\delta^{15}N$ varied across nests, with the highest values throughout the profile registered in DS-E5-N1 (average value ~22.5‰) and the lowest in DS-E2-N1 (average value ~3.8‰) (Fig 9). With the exception of DS-E2-N1 that showed minimal variation, pronounced declines in $\delta^{15}N$ were observed across nest profiles starting ca. 1835 in DS-E3-N1 (from ~11‰ to 4‰), ca. 1915 in DS-E1-N2 (from ~10‰ to 6‰), and ca. 1940 in DS-E5-N1 (from ~26‰ to 19.5‰) (Fig 9). Overall, trends in diatom compositional changes (PCA-1$_{diatoms}$) and shifts among the four siliceous indicators (PCA-1$_{indicators}$), varied across nests but generally underwent the greatest change after the early to mid-20th century (Fig 4), noting the shorter profile for DS-E1-N2 (Fig 8).

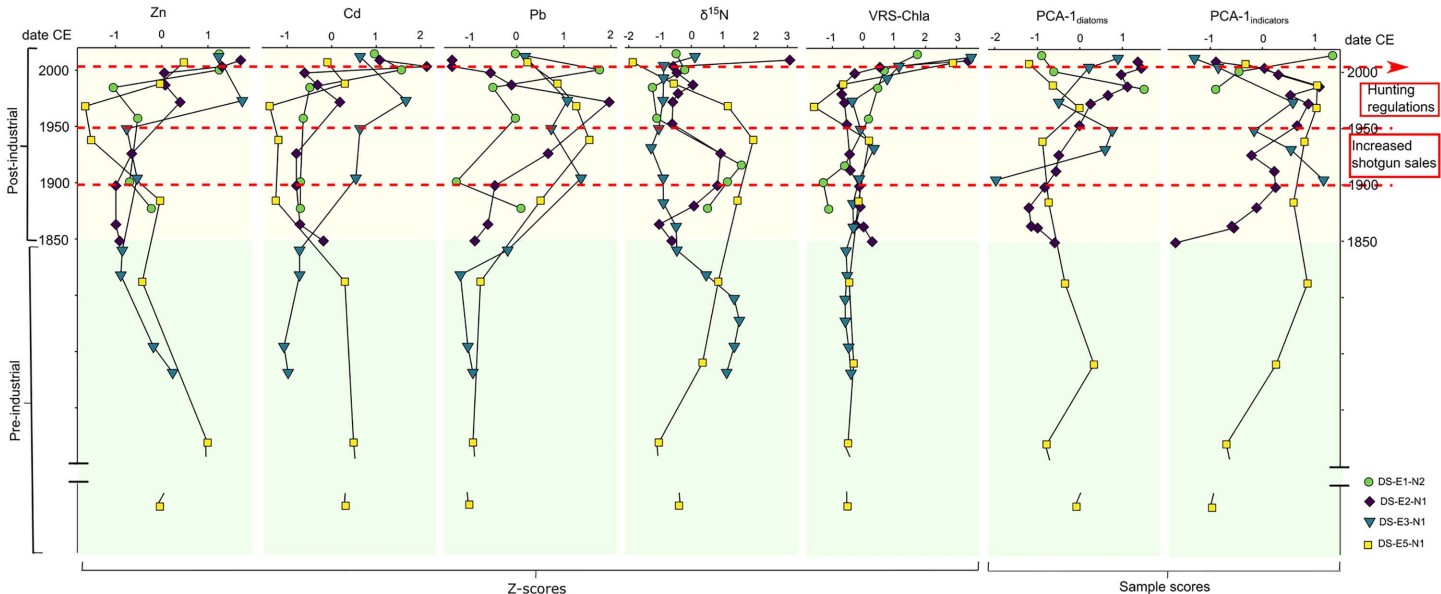

**Fig 8. Summary of the main ornithogenic proxies studied (expressed as Z-scores) and PCA axis 1 sample scores of diatoms and for all the siliceous indicators for the four eider nests.** $^{210}$Pb gamma dates are shown on the y-axis and are based on the constant rate of supply model. The yellow and green shading depict the pre-industrial and post-industrial eras, respectively. The horizontal red dotted line indicates the start of regulated hunting in Greenland. Note that the y-axis has been truncated to facilitate comparisons at similar time scales across nest profiles.

## Discussion

### Do eider nests accumulate stratigraphically to yield reliable chronologies and paleoecological archives?

Based on the four Digges Sound eider nest profiles examined in this pilot study, nests of philopatric sea ducks can be effective archives of paleoecological proxies. Among eider ducks, in which nests are not disturbed by abiotic conditions, predators, or by nesting eiders digging out a new nest bowl each year, the vegetation ring around the nest bowl accumulates material in a stratigraphic manner. Much like an aggrading peat sequence, this material would humify (process by which organic matter breaks down and transforms into humus, a dark, stable organic material) over time providing an ideal historical record to sample.

Given the generally diffuse $^{137}$Cs activity profiles and the absence of a clear peak, this isotope cannot be used to identify the 1963 height of global nuclear fallout in the eider nest profiles. The lack of a clear peak is likely a reflection of the organic and ombrotrophic peaty nature of the nest samples, as $^{137}$Cs under these conditions can be affected by post-depositional migration (both active uptake and downward diffusion) [66,67]. In addition, it is expected that there would be lower $^{137}$Cs activity in nests compared to lake sediment, given the much smaller surface area of the nests [35]. The $^{210}$Pb dating profiles from the four nest profiles generally followed the expected decline in activity with depth, an encouraging trend for establishing reliable chronologies. Although not included in our analysis, but of interest to highlight differences in nest ages, a very short nest profile, DS-E1-N1, was likely too young to yield a chronology as $^{210}$Pb activity remained high and did not decline over the 4.5-cm depth of the nest sequence (Fig 3e). Ombrotrophic sequences that depend solely on atmospheric inputs (such as eider nests) may be particularly well-suited for $^{210}$Pb dating when using the CRS [36], as this model assumes a constant supply of $^{210}$Pb to the surface of the aggrading material, allowing for temporal variations in accumulation rates. A potentially confounding secondary source of $^{210}$Pb to the nest may come from material delivered by eider ducks to the nest; however, this was discounted in other paleoecological seabird work in the Arctic [68] since the

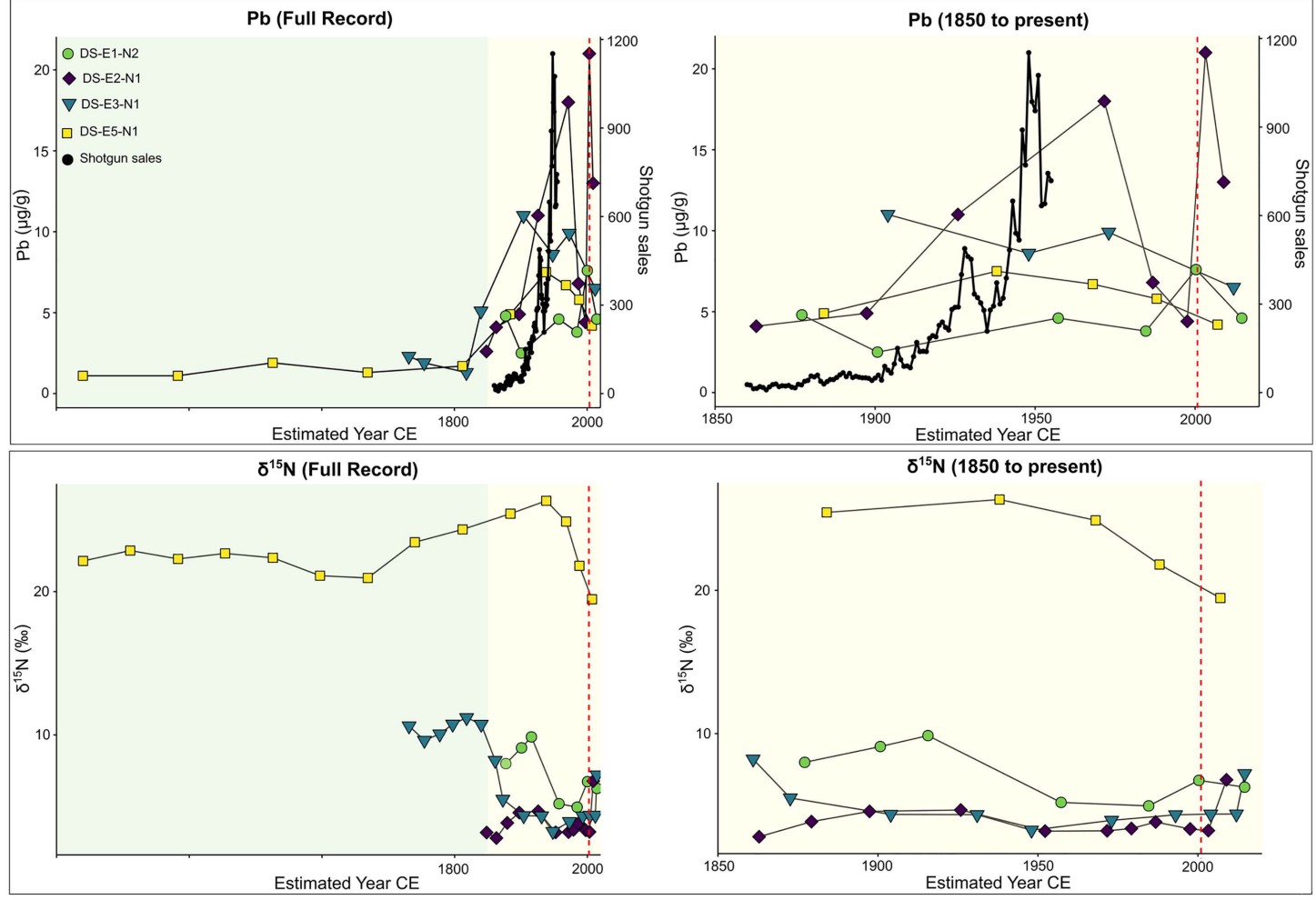

**Fig 9. Total stable Pb concentrations (μg/g) plotted against [210]Pb dates for the four Digges Sound eider nest profiles studied, compared to the recorded number of shotguns traded or sold in West Greenland between 1860 and 1955 (top panel; [62]).** Coloured areas distinguish the pre-industrial (green) and post-industrial (yellow) eras. The vertical red dotted line indicates the start of regulated hunting in Greenland. The figure on the right presents a zoomed-in view highlighting trends during the post-industrial era. Sedimentary δ15N (‰) plotted against [210]Pb dates for the four eider nest profiles are shown in the bottom panel. Coloured areas distinguish the pre-industrial and post-industrial eras. The vertical red dotted line indicates the start of regulated hunting in Greenland. The figure on the right presents a zoomed-in view highlighting trends during the post-industrial era.

fluxes observed at Cape Vera were similar to or even lower than those measured at other Arctic locations not influenced by seabirds. The [210]Pb dating results of our study demonstrate that eider nest material aggrades in a vertical manner and, if carefully and critically assessed, can yield reasonable chronologies (Fig 3).

The eider nest archives contained reasonable geochronological profiles, with relatively high surface [210]Pb activity (for Arctic regions) that generally decayed with depth. Our radioisotopic dating results suggest that nesting eiders have been occupying the islands of Digges Sound for at least the past ~780 years. Radiocarbon dating of nest profiles, lake sediment cores [1], and unpublished peat core data all support this conclusion. However, the variation in the duration of nesting occupancy detected (i.e., presence of biological proxies, changes in δ15N values and VRS-chla) suggests that determining the general timing of eider colony occupancy will require that a larger number of nests be sampled, particularly given that nest age varied despite their similar outward appearance.

Diatoms are especially useful for tracking changes in nutrient and moisture conditions within the nests, as shifts in species composition reflect variations in these environmental variables over time. In addition, chrysophyte stomatocysts, testate amoebae, and phytoliths provided complementary information on moisture variability and surrounding vegetation. The diatoms observed in the four nest sequences were almost exclusively taxa that have been commonly reported from low moisture soils or ephemeral environments, such as *Pinnularia borealis*, *Mayamaea atomus*, *Hantzschia amphioxys*, and *Luticola mutica* (e.g., [69–73]). Aerophilic taxa are expected in records from a dry, semi-aquatic environments, such as the peaty material comprising the eider nests. Given that eider females rarely leave their nests during the laying and incubation period and, when they do, it would be to drink water from nearby freshwater ponds [10,74], the eider guano produced during this nesting period could potentially contain diatom taxa from the lake. However, the aerophilic diatom assemblages observed in our nest profiles differed substantially from sedimentary diatom records collected from nearby ponds on these same Digges Sound islands [25,75]. In addition, there were no diatom remains observed in eider guano samples collected for this study, similar to the findings in Andean Condor (*Vultur gryphus*) guano reported by Duda et al. [76]. Based on these lines of evidence, aggrading eider nest material can provide sufficient moisture and nutrients to support the establishment and growth of *in situ* diatom communities and can yield paleoecological archives.

## Can nest archives track long-term changes in eider populations and their environment?

The Digges Sound eider nest records in this pilot study contained a rich variety of paleo-proxies that show promise for tracking nesting history over at least the past few hundred years. Nesting eiders are known to directly affect their environment through the addition of nutrient-rich guano that stimulates vegetation growth [2]. We have established that the aggrading nest materials can provide the resources (moisture and nutrients) required for the establishment of primary producers such as diatoms. In the relatively harsh semi-terrestrial growing conditions found in eider nest material, diatom taxa would be largely sustained by a sufficient supply of moisture and nutrients, similar to terrestrial assemblages observed in other aerophilic soil environments [73,77–80]. In these rather extreme environments, changes in these restricted resources would be expected to be associated with changes in eider populations and the surrounding environment. Stable nitrogen isotopes served as a direct proxy for nesting eider activity, providing insights into seabird-derived nitrogen release in coastal environments over space and time. Variations in δ¹⁵N measurements are a reliable indicator of seabird inputs, increasing by ~3.4‰ per trophic level [81]. Eiders contribute elevated δ¹⁵N levels through their feces, feathers, eggshells, and carcasses, providing valuable insights into historical nesting colonies. In contrast, subfossil diatoms offer a more indirect method, tracking changes in local nesting conditions influenced by environmental factors such as moisture.

Trends in biological proxies varied substantially among nests, highlighting that future studies should sample a larger number of nests to achieve a more comprehensive and robust assessment of past environmental conditions in nesting areas. Nevertheless, we hypothesize that the scarcity of siliceous indicators in the earlier sequences of two of our nests (DS-E3-N1 and the shorter profile of DS-E1-N2) indicated that conditions were not favourable at this time for the establishment of biological assemblages. For example, in DS-E3-N1, the lack of observable diatoms and other siliceous indicators in the pre-1900 nest sequences may suggest that conditions such as moisture and nutrients were insufficient to sustain biological populations during the colder period of the LIA [82]. Given that nutrients and moisture for diatom growth are supplied by the eiders when occupying the nest, it seems reasonable to suggest that this nest was not preferentially selected during this relatively severe climate period. Furthermore, a reduction in guano-derived nutrients is also consistent with the low stable nitrogen isotope values registered in this nest sequence during this period, further supporting irregular nest occupancy during this time. In contrast, the oldest nest (DS-E5-N1) provided evidence of nearly continuous occupation for several centuries, suggesting that this was a favoured nest and regularly selected by eiders. Here, siliceous indicators were in countable abundances for the entire record of this oldest nest sequence, suggesting that the continuous presence of eiders provided nutrients and moisture to sustain diatom growing conditions, even during the early period

(ca. 1240 to ca. 1850) when other nests lacked siliceous indicators. Further support for continuous occupation of this nest is the substantially higher $\delta^{15}N$ values observed throughout the DS-E5-N1 profile relative to our other nest sequences (Fig 9), consistent with a sufficient supply of nutrients (and moisture) to promote the establishment of diatoms and other indicators at this time. High relative abundances of diatoms commonly found in arid terrestrial environments such as low moisture soils (*P. intermedia, L. mutica* and *M. atomus*) and the high relative abundance of phytoliths (likely from grasses and sedges) at the early stages of this nest profile suggests that, although a relatively dry or ephemeral environment, eiders supplied sufficient moisture and nutrients for the development of a vegetation ring around the nest. DS-E2-N1 also registered countable numbers of biological proxies throughout the length of the record but is a younger nest that was only established ca. 1850, after the LIA era (Fig 5; [82]).

Elevated metal concentrations in regions with high eider activities have been observed in freshwater pond sediments from Hudson Strait [59] and in sediment records from a lake proximal to a thick-billed murre (*Uria lomvia*) colony on Digges Sound [24]. These findings highlight the importance of using these elements as proxies of seabird population dynamics. In this study, the analysis of metals in eider nests was essential for understanding long-term trends in this species' nesting behavior and ecological influence. The higher concentrations of Zn and Cd in nest DS-E5-N1 during the pre-industrial era relative to our other nest sequences may be another indication that this oldest nest of our study has been continuously occupied for centuries by eiders.

Warmer conditions at the end of the LIA (ca. 1850) generally corresponds to the beginning of the industrial era, marked by greater global outputs of materials from anthropogenic activity, which activity even extended to remote sub-Arctic regions [83,84]. In addition, these contaminant signatures are well-preserved in paleo-records [83,85] offering valuable insights about historical pollutant deposition. Concurrently, this era was characterized by dramatic advancements in hunting technologies such as the introduction and increased availability of firearms, ammunition, and motorized boats, all of which revolutionized eider harvesting practices in west Greenland and Atlantic Canada [1]. In fact, the use of lead (Pb) shot in hunting has been a significant threat to waterfowl due to lead poisoning from accidental ingestion, which has led to population declines in some cases [86,87]. In addition to these anthropogenic influences, climate warming also modifies eider populations by altering their nesting conditions, potentially expanding suitable habitats and resources but also increasing predation pressure [88].

Post-LIA climate amelioration in the Canadian Arctic starting in the mid-19th century [89–91] would have resulted in improved environmental conditions for diatom and vegetation growth within and around the eider nests. Diatoms and other siliceous indicators may indicate warmer conditions after the LIA [89,90] in all the profiles except for DS-E1-N2, where indicators remained uncountable until 1980. The lack of countable biological indicators in this one nest may be due to a lack of nutrients. Further studies based on a higher number of nests to better interpret paleoecological records are needed. For example, with the exception of DS-E5-N1, there were notable changes in the diatom records at the turn of the 20th century (e.g., the onset of countable diatoms in DS-E3-N1 ca. 1882) that coincided with warmer conditions following the LIA (Fig 6). Although more difficult to interpret as a warming signal, nest DS-E2-N1 (Fig 5) registered a pronounced compositional shift ca. 1890 from an assemblage dominated by *P. viridiformis* and *Stauroneis gracilis* to an assemblage dominated by other *Pinnularia* taxa (*P. intermedia* and *P. borealis*). The prevalence of these *Pinnularia* species may indicate a shift in habitat conditions, with an increase in nutrient inputs together with drier conditions. *S. gracilis* is widely distributed in aquatic ecosystems, but particularly favours oligotrophic environments characterized by good water quality [77,92]. Similarly, *P. viridiformis* has been observed in habitats with low nutrient inputs [93,94], whereas, *P. borealis* is a highly stress-tolerant aerial diatom, present even in extreme environments [73,95]. Meanwhile, *P. intermedia* shows notable adaptability, occurring in moderately nutrient-rich settings and disturbed dune environments, so is able to persist in varying ecological conditions [80]. In the case of DS-E5-N1, phytolith abundance reached its lowest levels, while protozoan plates peaked during this period. This is consistent with drier conditions associated with a warmer environment during this period: phytoliths are correlated to temperature and

humidity, showing higher production in moist environments [96], whereas protozoan plates have been associated with drier conditions, as observed in peatlands [97].

The introduction and the uptake of modern hunting techniques ca. 1860 undoubtedly had negative consequences on nesting eider population size on the islands of Digges Sound [1,98]. The declining size of local eider populations during the industrial era was evidenced by a decline in δ15N values starting ca. 1900 registered in most of our nest profiles. While increased predation or disease outbreaks could have led to similar results, we have no evidence of this from Inuit local ecological knowledge [16] or any other source. However, the decline in δ15N values registered in our nest profiles matches inversely to the increasing shotgun sales trend in Greenland (Figs 8 and 9), where most eastern sub-Arctic Canadian nesting eiders overwinter (e.g., [99]). The post-industrial period is characterized by an increase in shotgun sales for Greenland that generally matches the post-1850 increase in Pb concentrations in all nest profiles (Fig 9). The increase in Pb concentrations registered in nest material during this time may be a result of either the ingestion of spent Pb shot by eiders [100] or from eiders carrying embedded Pb shot [101–103]. Although local hunting pressure was lower in this area than in regions with larger eider numbers, Hargan et al. [1] suggested that declines in the eider population over the 20th century may be linked to an increase in Greenland's human population and in the availability of firearms [21,98]. Increased access to motorized boats [1] may further explain the increase in Pb concentration in our eider nests, as this would have facilitated greater access to more offshore hunting locations particularly in poor weather conditions that would have extended the hunting season in winter. Importantly, leaded gasoline began to be used extensively in the 1920s and this would have coincided with increased atmospheric deposition of anthropogenic Pb [104], including to remote places like Digges Sound [105,106]. Thus, the increase in post-industrial nest Pb concentrations may reflect both the increased use of Pb shot ammunition and the concomitant increase in atmospheric deposition of Pb.

In the mid- to late-20th century, our multiple nest proxies registered distinct changes in nest conditions that we speculate are ultimately linked to a shift in eider population dynamics (discussed below). For example, a decrease in Arctic ice cover, rising temperatures, and longer growing seasons over the past ~50 years could promote increases in eider clutch sizes and earlier nest initiation dates [88], as has been noted in populations elsewhere [107]. Additionally, the 1994 update to the Canadian Migratory Birds Convention Act, alongside Greenland's hunting regulations, granted enhanced protection to eiders, reducing hunting pressure in the migration and wintering areas where eiders from Digges Sound reside and over-winter [10]. Collectively, these changes likely contributed to eider population growth during this time. Evidence of a four-fold increase in the eider nesting population during this time has been reported in the Digges Sound region based on the observed increase in seabird numbers at several eider nesting sites in the area [1,13]. Likewise, Merkel [108] observed an increase in Greenland eider populations following the implementation of stricter hunting regulations in Greenland ca. 2000, with breeding numbers increasing by 212%, from 2000 to 2007.

Indeed, the biological proxies in many of our nest profiles register notable changes that may be indicative of improved growing conditions such as increased moisture and greater nutrient availability, both of which could be attributed to a more regular occupancy of the nest bowls. Changes in diatom and other siliceous indicators may indicate changes in nutrient and moisture availability during the 20th century, which may be due to the presence of eiders, warmer conditions, or both (Figs 4–7). One of the most conspicuous examples of this change is registered in nest DS-E1-N2 (our shortest diatom record), where diatoms became countable for the first time in the mid- to late-20th century. We speculate that the lack of biological proxies in the earlier parts of this record may indicate that this nest was not as desirable as others in our survey (due to location, lack of nesting material, or site unfamiliarity). Responses to 20th century warming were evident across all nests, marked by significant shifts among taxa. For instance, subaerial diatoms transitioned to taxa associated with nutrient-rich environments, such as *H. amphioxys* [109], or wetter conditions, like *N. inconspicua* [110]. In addition, the increase in phytolith and diatom total abundances during this period further support the signal of higher humidity and nutrient availability promoting their growth [96,111].

Concerning the metal(loid)s studied, the general increase observed especially in Zn and Cd concentrations, previously recognized as ornithological tracers [24,59], together with the general increase in δ15N, likely derived from 15N-enriched eider guano as suggested by Hargan et al. [1], and VRS-chla in the latter part of the 20th century, may further support increased and/or more consistent nest occupation by eiders (Fig 8). Zn and Cd generally increased in all the nests after ca. 1950, with the increase in Cd more striking in DS-E2-N1. The shift to a nutrient-rich environment during this period (ca. 1950-ca. 2000), attributed to an increased presence of eiders in the nests, is reinforced by the notable increase in VRS-chla in all the nest profiles. Lastly, the ban of lead shot for migratory bird hunting in Canada [1] together with the ban of leaded gasoline in Canada in 1990 [112] may have led to the declining Pb concentrations in nests detected after ca. 2000 (Figs 9 and S1), although more data would be required to provide stronger conclusions.

## Eider nest archives: Challenges, future directions and conclusions

Our pilot study showed promising results that may be useful for ornithologists and conservation biologists in general. The application of paleoecological approaches to nest archives can inform conservationists about historical shifts in environmental conditions and bird population dynamics, offering a baseline to evaluate the effectiveness of current conservation measures. The historical resilience of eiders is essential for conservation strategies (e.g., [108]) by ensuring the protection of nesting areas that might be affected by potential natural and/or anthropogenic threats. Similarly, Duda et al. [76] highlighted the importance of long-term nesting site data in their study of the Andean Condor. Our findings demonstrate that eider nesting materials accumulate vertically over time and yield reasonable dating profiles. The nests examined from Digges Sound archived different paleoecological proxies providing information about the history of bird populations. We suggest that given the variation detected, future studies should aim to sample a greater number of nests in order to generate more consistent findings and clearer interpretations of bird populations and colony occupancy. These data can reveal trends in factors that may be linked with broader ecological shifts such as climate change or human disturbances.

Our research provides a promising foundation for further research on other philopatric, ground nesting seabird and sea duck species that inhabit remote locations where data are scarce. Combining nest archive data with other monitoring tools (e.g., colony surveys, acoustic monitoring, remote sensing, and Indigenous knowledge), and incorporating additional records from nearby habitats, such as ponds or lakes as well as other proxies (i.e., fecal sterols and stanols; [7]), would also help to validate findings and provide a more comprehensive view of seabird and sea duck ecology. Additionally, by highlighting environmental conditions as well as human activities (e.g., hunting) linked to the increase or decrease in seabird abundances as we have done, nest archives can guide habitat protection priorities and the development of adaptive management strategies. Nonetheless, we acknowledge that larger sample sizes will be needed to provide more robust conclusions.

## Supporting information

**S1 Fig. Selected element concentrations plotted against 210Pb dates for DS-E1-N2 (green); DS-E2-N1 (purple); DS-E3-N1 (blue) and DS-E5-N1 (yellow).** Elemental concentrations are either presented in μg g−1 or mg g−1 dry weight as stated.
(TIF)

**S1 Table. 14C chronologies of the nest profiles analyzed (DS-E3-N1 and DS-E5-N1).**
(PDF)

**S2 Table. List of diatom species mentioned in this study along with their taxonomic authorities.**
(PDF)

**S1 File. Supplementary text for *Exploring the potential of nest archives for establishing long-term trends in local populations of an Arctic-nesting colonial sea duck* manuscript.**
(DOCX)

## Acknowledgments

The authors would like to thank Michael Janssen and Jake Russell-Mercier (ECCC) and Chris Grooms (Queen's University) for logistical support, and the Ivujivik Hunters and Trappers Association for their support in our research and fieldwork assistance.

## Author contributions

**Conceptualization:** Inmaculada Álvarez-Manzaneda, Kathleen M. Rühland, Mark L. Mallory, Kathryn Hargan, John P. Smol.

**Formal analysis:** Inmaculada Álvarez-Manzaneda, Kathleen M. Rühland, Marlo Campbell, Kathryn Hargan.

**Funding acquisition:** Inmaculada Álvarez-Manzaneda, H. Grant Gilchrist, John P. Smol.

**Investigation:** Inmaculada Álvarez-Manzaneda, Kathleen M. Rühland, Marlo Campbell, Nik Clyde, H. Grant Gilchrist, Kathryn Hargan.

**Methodology:** Inmaculada Álvarez-Manzaneda, Kathleen M. Rühland, Matthew P. Duda, Kathryn Hargan, John P. Smol.

**Project administration:** Kathleen M. Rühland, Nik Clyde, Kathryn Hargan, John P. Smol.

**Resources:** H. Grant Gilchrist, Kathryn Hargan, John P. Smol.

**Supervision:** Kathleen M. Rühland, H. Grant Gilchrist, Kathryn Hargan, John P. Smol.

**Validation:** John P. Smol.

**Visualization:** Inmaculada Álvarez-Manzaneda, Kathleen M. Rühland, Marlo Campbell, Kathryn Hargan.

**Writing – original draft:** Inmaculada Álvarez-Manzaneda, Kathleen M. Rühland.

**Writing – review & editing:** Kathleen M. Rühland, Marlo Campbell, Matthew P. Duda, Mark L. Mallory, Nik Clyde, Kathryn Hargan, John P. Smol.

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
