## [Decision Letter · Decision Letter 0]

15 Apr 2025

Dear Dr. Hargan,

We look forward to receiving your revised manuscript.

Kind regards,

Simon Belle, Ph.D.

Academic Editor

PLOS ONE

Journal Requirements:

2. Thank you for stating the following financial disclosure: [This research was funded by Environment and Climate Change Canada, the W. Garfield Weston Foundation, a Kenneth M. Molson Foundation grant, the Natural Sciences and Engineering Research Council of Canada (NSERC), Oceans North and the PEW Charitable Trusts, Nunavut General Monitoring Plan, ArcticNet Network Centres of Excellence, and was within the project PAST, financially supported by the European Commission (H2020-MSCA-IF-2019, Grant no. 897535).]. 

3. In the online submission form, you indicated that your data will be submitted to a repository upon acceptance. We strongly recommend all authors deposit their data before acceptance, as the process can be lengthy and hold up publication timelines. Please note that, though access restrictions are acceptable now, your entire minimal dataset will need to be made freely accessible if your manuscript is accepted for publication. This policy applies to all data except where public deposition would breach compliance with the protocol approved by your research ethics board. If you are unable to adhere to our open data policy, please kindly revise your statement to explain your reasoning and we will seek the editor's input on an exemption.

4. We note that Figure 2 in your submission contain [map/satellite] images which may be copyrighted. All PLOS content is published under the Creative Commons Attribution License (CC BY 4.0), which means that the manuscript, images, and Supporting Information files will be freely available online, and any third party is permitted to access, download, copy, distribute, and use these materials in any way, even commercially, with proper attribution. For these reasons, we cannot publish previously copyrighted maps or satellite images created using proprietary data, such as Google software (Google Maps, Street View, and Earth). For more information, see our copyright guidelines: http://journals.plos.org/plosone/s/licenses-and-copyright.

1. You may seek permission from the original copyright holder of Figure 2 to publish the content specifically under the CC BY 4.0 license.  

Additional Editor Comments:

Dear Authors,

Your manuscript PONE-D-25-09098 "Exploring the potential of nest archives for establishing long-term trends in local populations of an Arctic-nesting colonial sea duck" has now been seen by two external reviewers whose comments are listed at the end of these lines.

The reviews were generally favorable, and reviewers found the study interesting because it numerous potential applications for conservation and management. Both reviewers provided some suggestions of edits that could make the manuscript even stronger. However, Reviewer 1 pointed out a couple major issues that need to be carefully considered: (i) very limited sample size, which restricts the strength of the conclusions; (ii) speculative elements in the discussion section; (iii) dependence on personal communications; and (iv) insufficient detail and clarity regarding nest dating which is a critical component of the study.

I further agree with reviewer 1 that the manuscript cannot be considered for publication in Plos One in such a form. A reliable chronology is essential in paleoecological research, especially when it comes to “exploring the potential of a new environmental archive”. All steps (method, result and discussion) must be presented in a precise and consistent way to improve transparency and reader´s understanding the robustness of the chronology. Due to the very limited sample size (only 4 nests, each with distinct histories), the main goal cannot be reliably addressed and the conclusions are rather speculative. The manuscript should be revised to reflect a narrower scope, by presenting it as a case study rather than a broadly generalizable analysis. While we understand that expanding the dataset may not be feasible at this stage, adjusting the study’s objectives accordingly will help align expectations.

When preparing your revised manuscript, you are asked to carefully consider the reviewer comments which are attached, and submit a list of responses to the comments. We look forward to receiving your revised manuscript.

We look forward to receiving a new version of your manuscript.

With kind regards,

Reviewers' comments:

Reviewer's Responses to Questions

**Comments to the Author**

1. Is the manuscript technically sound, and do the data support the conclusions?

Reviewer #1: Partly

Reviewer #2: Yes

2. Has the statistical analysis been performed appropriately and rigorously?

Reviewer #1: Yes

Reviewer #2: Yes

3. Have the authors made all data underlying the findings in their manuscript fully available?

Reviewer #1: No

Reviewer #2: Yes

4. Is the manuscript presented in an intelligible fashion and written in standard English?

Reviewer #1: Yes

Reviewer #2: Yes

Reviewer #1: Review of PONE-D-25-09098

In this manuscript, the authors assess the historic nest use of Northern Common Eiders (*Somateria mollissima* borealis) on four islands in Digges Sound, which are commonly used for breeding and chick-rearing by the arctic sea ducks. Because this species has been monitored only sparsely in the past, the authors apply various paleolimnological techniques, including stable nitrogen isotopes (δ^15N), visible reflectance spectroscopy-inferred chlorophyll a (VRS-chla), and siliceous microfossils such as chrysophyte cysts, phytoliths, and protozoan plates, in addition to diatoms and metals from cores taken of nests to obtain proxies for historic nest use by eiders. The authors test whether these proxies can give reliable information on population island use by a species known to reuse nests. The chemical and organismal signatures they sampled and analyzed from the stratospheric were thorough. To further verify their chemical findings, the authors provide additional context by discussing the Little Ice Age and novel anthropogenic pressures such as increased hunting efficiency due to shotguns and motorized boats beginning around 1860, which they linked closely to fluctuations in the eider population proxies they studied. They found that by combining paleoecological data with anthropogenic and climate history, they could predict eider nest use dating back to the 13th century with the caveat that challenges with radiocarbon dating introduced multiple potential dates, highlighting the complexity of dating these archives and emphasizing the need for caution when interpreting temporal results. The authors highlight that the nest samples in their study closely align with similar findings in peat sequences in that the nests stratify vertically over time and thus provide an excellent paleoecological archive.

The authors do an excellent job justifying the necessity of developing reliable proxies to monitor populations and their chosen focal species. Their methods and statistics are sound, and they used a variety of approaches in tandem to ensure their findings could reliably link to Common Eider population presence. However, there are two main issues with the manuscript. The sample size is too small, and the interpretation of their findings along with their study question are much too broad in scope compared to what the study accomplished. With a sample size of four nests, the authors claim this study provides insight into species-wide eider nesting behavior and population dynamics. Four nests are simply insufficient to provide information on these species and population scale questions, as the patterns they found may not accurately represent the species. This is highlighted by the fact that the four nests they analyzed revealed very different histories. Far more nests would need to be sampled to find large-scale patterns. In addition, the temporal scale of their findings is limited. They dated one nest back to the 13th century, two to the 19th century, and one could only be dated to the 1980s. In the introduction, the authors justify their methods by saying we do not understand eider population dynamics pre-human presence in the Canadian sub-arctic. However, they were not able to date back to this time. Though one nest was dated to before the Industrial Revolution, they would need more nests to address the questions they present in the introduction. As the manuscript is presently, the authors leave much of the discussion as inherently speculative.

In conclusion, the presented study is an intriguing method to address a problem in assessing vitally important historical population trends that could have lasting conservation impacts. The study offers a valuable preliminary investigation into this method, and the authors could strengthen future work by expanding their sample size. The authors need to re-frame the manuscript before publication by narrowing their interpretive scope.

Major Comments:

Introduction line 115– Objective (iii) – “..track long-term changes in the nest environment and populations.” is not covered in the study design or sample size. I suggest removing this goal entirely or mentioning that this was initially a goal that could not be addressed by the limited sample size.

Materials and Methods lines 162-172 – Somewhere in this paragraph an explanation for the limited sample size should be included.

Materials and Methods – The authors should justify their choice of using CONISS and broken-stick analysis in their methodology, given that PLoS ONE readers come from diverse fields and may not be familiar with these approaches. Briefly explain why these methods were selected and how they are specifically relevant to analyzing diatom assemblages in paleoecological contexts.

Radioisotopic dating (210Pb and 14C) – This section would benefit from more explicit justification and explanation to make the methods more accessible to a broader readership. Specifically, the authors should clarify why 210Pb and 14C dating were chosen and justify using the CRS model for 210Pb dating, explicitly outlining its suitability and limitations for this type of sample. Furthermore, additional details on how intervals for dating were selected, especially given the stated difficulty in interpreting specific radiocarbon dates due to multiple possible age outcomes, would improve transparency and comprehension.

Geochemical analyses – This section would benefit from further justification and detail to enhance clarity for the broader readership. The authors should expand on why specifically Zn, Cd, and Pb were selected for analysis, particularly if their utility as proxies for seabird inputs is widely established—providing more than one citation here would strengthen this claim. Additionally, the ‘historical shotgun sales data’ mentioned requires explicit referencing: please clearly state the source of this data and the geographic region the data encompasses, describe briefly how this dataset was obtained, and clarify any procedures used to clean or prepare these data for analysis.

Diatoms and other siliceous indicators – This section could be strengthened by clarifying several methodological choices to make the manuscript more accessible to readers unfamiliar with paleoecological techniques. Specifically, please briefly justify why samples were collected at 1-cm intervals—was this resolution chosen to capture sufficient ecological detail, and could other resolutions have offered different insights? Additionally, clarify the exact range or criteria determining how many times the rinsing procedure was repeated before reaching neutrality. The authors should also explicitly state the method used for diatom identification—morphological or genetic—and, if morphological identification was employed, reference the key literature or databases utilized. Lastly, explain why counting a minimum of 400 diatom valves per interval was selected, and briefly justify any deviations from this count in samples with fewer diatoms present.

Results: Radioisotopic dating (210Pb and 14C) – For nest DS-E3-N1, the authors note they could not reliably date the bottom-most interval due to poor-quality material. They subsequently mention using the R package ‘clam’ to generate the best-fit date for a higher interval. However, the authors do not specify precisely how the model accounted for this adjustment or missing data. Please clearly describe what assumptions or methods (e.g., extrapolation or interpolation) were used within ‘clam’ to estimate the date of this interval. Clarifying this step is vital for ensuring readers understand the robustness of the generated chronology.

Results and Discussion – The authors rely on personal communications (pers. comm.) several times in the results and discussion sections, particularly when discussing the absence of diatoms in eider guano. While personal communications can occasionally be appropriate to supplement published evidence, they are insufficient as standalone evidence to support significant interpretations or conclusions. Please either provide published studies or empirical data supporting these statements or clarify clearly why published data were not available. Minimizing dependence on personal communications for critical aspects of the interpretation would strengthen the manuscript and improve transparency.

Discussion – The authors mention in the Results that "no clear ¹³⁷Cs peaks were detected." Given that ¹³⁷Cs is commonly used as an independent chronological marker (particularly around the period of nuclear fallout), the absence of detectable peaks needs explicit interpretation. In the Discussion, please explain why ¹³⁷Cs peaks were not found in these samples. For instance, could the absence reflect local conditions, methodological issues, the nature of the nest material, or limited atmospheric deposition at this site?

Discussion – On line 478 and elsewhere, the authors discuss nutrient and moisture availability as potentially insufficient for biological assemblages during the Little Ice Age based primarily on limited data from two nests. However, they omit discussion of the fact that nest DS-E5-N1 was established well before this period, indicating at least localized continuous nesting and sufficient nutrients and moisture for biological proxies to persist throughout the Little Ice Age. Given the small sample size (four nests) and conflicting evidence from DS-E5-N1, this interpretation about nutrient availability during the Little Ice Age should be clearly marked as speculative. It would be more appropriate to explicitly acknowledge this limitation, clarifying that a larger sample size would be necessary to confidently address eider occupancy and nesting conditions during this period.

Overall manuscript – The sentence at line 473 in the Discussion very clearly and effectively articulates the core rationale and intended scope of the study. I recommend the authors consistently emphasize and reiterate this perspective throughout the manuscript—particularly in the Introduction, Results, and Conclusions sections—to align their stated goals with the scope of their findings. Reinforcing this point more explicitly throughout the manuscript will further clarify the study’s contributions and limitations.

Figures – The quality of several figures, particularly Figure 8, needs improvement to enhance readability and accessibility. Currently, the resolution of all figures is low, making text and symbols difficult to discern. In addition, the color choices in Figure 8 are challenging to distinguish, especially for readers with color vision deficiencies. Consider using a colorblind-friendly color palette.

Figure 9 – This figure currently displays a timeline spanning over 800 years, making it very challenging to interpret trends occurring around the introduction of regulated hunting due to compressed visual resolution. To enhance clarity and emphasize patterns most relevant to their hypotheses about regulated hunting impacts, the authors should consider dividing this figure into multiple panels and/or providing an inset zoomed specifically around the post-industrial era and following the introduction of regulated hunting. This visual restructuring would significantly improve interpretability, highlighting key temporal changes related to anthropogenic impacts.

Minor Comments:

Line 55 – The authors introduce the economic and cultural significance here, yet they do not explain this significance until line 68. Remove this line or move it to line 68.

Line 85 – The word ‘local’ is stated twice. I suggest removing one.

Digges Sound: site description and brief history of the region lines 129-132 – Mention specifically any help received by locals, especially the Ivujivik community.

Figure 2 – Add provincial and state borders to further clarify geographic location.

Lines 142-145 – Cite nest selection behavior.

Lines 142-145 – Add a sentence about the frequency of new bowl selection by eider females (i.e., how often to females start an entirely new nest?); cite.

Materials and Methods lines 152-172 – I suggest specifying when eider breeding occurs and whether or not data collection overlapped with the breeding season.

Materials and Methods line 162 – Add sample size (e.g. n = 4) and briefly mention one nest was not used.

Line 172 – Justify 4°C

Chlorophyll a line 249 – Specify which variables.

Stable nitrogen isotopes line 266 – It would be beneficial for the authors to briefly clarify the meaning of "AIR" as the reference standard for δ¹⁵N measurements, specifying explicitly that it refers to atmospheric nitrogen.

Lines 268-271 – ..”to facilitate comparisons..” is used two sentences in a row. I suggest rephrasing one of the two sentences.

Results Line 298 – The authors should briefly explain or define "CE" (Common Era) upon first mention in the manuscript. A simple parenthetical clarification such as "(CE, Common Era)" would enhance readability.

Results Line 298 – Define “cal BP” upon first use.

Results line 368 – Justify use of CONISS and broken stick analysis; briefly explain these methods in the context of the field.

Discussion line 478 – Briefly define "humify" or "humification" to improve readability for non-specialists unfamiliar with soil and peat-forming processes.

Discussion line 487 – CRS is already defined in the materials and methods. I recommend writing the complete phrase ‘constant rate of supply model’ each time, as it is infrequently used. Alternatively, use CRS here without definition since it has already been defined.

Discussion line 500 – Clarify which specific result detected the variation in the duration of nesting occupancy.

Discussion lines 571-574 – This sentence is especially clear and effectively contextualizes the results within the broader literature. This type of explicit connection to past studies greatly enhances the reader’s understanding and interpretation of the findings and adds valuable additional context to the observed results in this manuscript.

Discussion lines 576-577 – The authors do not have enough data to claim that they, “reconstruct[ed] long-term trends in this species’ nesting behavior and ecological influence.” I strongly suggest rephrasing or removing.

Discussion line 580 – Spell out “Little Ice Age.”

Discussion line 634 – I suggest substituting the use of the informal ‘and/or’ for “may be a result of either the ingestion of spent Pb..or from eiders carrying embedded Pb shot.”

Discussion line 642 – A period is missing.

References – Address inconsistencies in journal name abbreviations and article title capitalization schemes. For example, Conservation Biology is abbreviated in reference 5 but not in reference 3. Some article titles are sentence case while others are in title case.

Reviewer #2: PONE-D-25-09098

The authors used paleoecological techniques to investigate historical usage indicators of Common Eider nests which can provide insight into historical ecological conditions, frequency of use, age, and population sizes. This pilot study is novel and innovative and deserves to be added to the body of literature. Potential applications for conservation and management of this culturally and ecologically valuable species are promising. I have a single minor suggestion, and found 2 typos. Overall, well-written and easy to follow.

General comment: It would be nice to have a short section explaining what the different groups/indicators/taxa signify ecologically and have a quick reminder at the beginning of the discussion for readers (like myself) who are unfamiliar with diatoms and siliceous indicators.

Line 342: unnecessary period following genus name (typo)

Line 642: Missing period at end of sentence

**Do you want your identity to be public for this peer review?** For information about this choice, including consent withdrawal, please see our Privacy Policy

Reviewer #1: **Yes: ** Joanna L. Corimanya

Reviewer #2: No

---

## [Author Response · Author response to Decision Letter 1]

2 Jun 2025

An attachment has been uploaded the addresses each reviewer and editor comment.

---

## [Decision Letter · Decision Letter 1]

3 Sep 2025

Exploring the potential of nest archives for establishing long-term trends in local populations of an Arctic-nesting colonial sea duck

PONE-D-25-09098R1

Dear Dr. Hargan,

We’re pleased to inform you that your manuscript has been judged scientifically suitable for publication and will be formally accepted for publication once it meets all outstanding technical requirements.

Kind regards,

Nicolas Gauthier, Ph.D.

Academic Editor

PLOS ONE

Additional Editor Comments (optional):

The reviewers have expressed satisfaction with the revisions you have made in response to their previous comments. The improvements to the manuscript's framing and presentation have successfully addressed the concerns raised during the initial review process. This is a strong, novel, and clearly articulated contribution to the field.

Reviewers' comments:

Reviewer's Responses to Questions

**Comments to the Author**

Reviewer #1: All comments have been addressed

Reviewer #3: (No Response)

2. Is the manuscript technically sound, and do the data support the conclusions?

Reviewer #1: Yes

Reviewer #3: Yes

3. Has the statistical analysis been performed appropriately and rigorously?

Reviewer #1: Yes

Reviewer #3: Yes

4. Have the authors made all data underlying the findings in their manuscript fully available?

Reviewer #1: Yes

Reviewer #3: Yes

5. Is the manuscript presented in an intelligible fashion and written in standard English?

Reviewer #1: Yes

Reviewer #3: Yes

Reviewer #1: (No Response)

Reviewer #3: In the paper "Exploring the potential of nest archives for establishing long-term trends in local populations of an Arctic-nesting colonial sea duck" the authors use solid paleolimnological techniques to bird nests in order to test the suitability of these techniques for infering the history of bird colonies. The techniques are used correctly and the results are very interesting.

This study is in my kowledge a completely new application and, as noted by the reviewers, very innovative so that its publication may be very useful, given its potential applications for conservation and management of this species.

The small sampling set for this pilot study is justified by the goal to minimize the disturbance to the colonies while testing the applicability of the techniques.

The paper is very well written and simple to follow.

The authors fully answered to the qiestions of the previous reviewers, so that I have no suggestion to give, and I propose to accept the manuscript as it is.

**Do you want your identity to be public for this peer review?** For information about this choice, including consent withdrawal, please see our Privacy Policy

Reviewer #1: **Yes: ** Joanna L. Corimanya

Reviewer #3: No

---

## [Editor Report · Acceptance letter]

PONE-D-25-09098R1

PLOS ONE

Dear Dr. Hargan,

I'm pleased to inform you that your manuscript has been deemed suitable for publication in PLOS ONE. Congratulations! Your manuscript is now being handed over to our production team.

Kind regards,

on behalf of

Dr. Nicolas Gauthier

Academic Editor

PLOS ONE